# Career Prospects of Young Dentists in Switzerland

**DOI:** 10.3390/ijerph17124310

**Published:** 2020-06-16

**Authors:** Guglielmo Campus, Philippe Rusca, Christine Amrhein, Andreas Meier, Oliver Zeyer, Thomas Gerhard Wolf

**Affiliations:** 1Department of Restorative, Preventive and Pediatric Dentistry, School of Dental Medicine, University of Bern, CH-3010 Bern, Switzerland; ph.rusca@sunrise.ch (P.R.); thomas.wolf@zmk.unibe.ch (T.G.W.); 2Department of Surgery, Microsurgery and Medicine Sciences, School of Dentistry, University of Sassari, I-07100 Sassari, Italy; 3Swiss Dental Association (SSO), 3011 Bern, Switzerland; ca@mund-werk.ch (C.A.); am@zahnarzt-meier.com (A.M.); oliver.zeyer@sso.ch (O.Z.); 4Department of Periodontology and Operative Dentistry, University Medical Center of the Johannes Gutenberg-University Mainz, 55131 Mainz, Germany

**Keywords:** career prospects, expectations, liberal dental practice, Switzerland, young dentists

## Abstract

The observational cross-sectional study was aimed to obtain information on the promotion and development of young professionals in Switzerland. An online survey with 20 questions was sent out. Data was collected on participants’ demographic data, including age, gender, level of qualification, place of work, information on employment, future perspectives, and career prospects. The survey was sent out to 1920 practitioners, of which 440 (22.9%) responded (37.1% males and 62.9% females). Of them, 76.6% were members of the Swiss Dental Association (SSO) 15.9% students, and 7.5% non-SSO members. Most participants had parents with a dental education (80.9%), and 19.8% did not. Young dentists in Switzerland most often saw their career prospects as neutral (39.8%) or rather positive (39.3%). Whereas significantly fewer dentists had a negative view of their professional future (16.8%), including more women than men, the fewest dentists of both sexes (4.1%) saw their career prospects as positive by far. The majority of young dentists were satisfied with their career prospects. Within the limitations of the current study, the reasons for this need further investigation. Despite good career prospects, there is a desire among young colleagues for cantonal practice assistance and mentoring programs, as well as support in finding a job and in taking the plunge into self-employment.

## 1. Introduction

Numerous forms of dental practice in Europe are possible. In many European countries, a trend toward group practices among dentists can be observed [1]. The ongoing changes of the profession often raise issues on the political agenda to develop solution strategies to support decisionmakers. To date, the careers of young professionals in Western countries could usually be discerned by reports of establishment of sale of practices and registers of employed professionals. A change in choosing the form of dental practice can also be seen in Switzerland. In a 2015 survey of dentists conducted by the Swiss Dental Association (SSO), the most common form of professional practice was single practice, with the majority stating that they worked in small villages with 2000 to 10,000 inhabitants [2]. More than a quarter of the sample declared to work in a group practice, and only 1% worked in either a dental center or a public institution [1,2]. The gender ratio of new dentists is now 0.4/0.6 with a predominant number of women. Students are often busy looking for a suitable job toward the end of their studies. In recent years, young dentist requirements have changed severely both in terms of professional aspects and the general conditions. Threats like the immigration of dentists from other countries, new investors in the oral health sector that start to open new dental centers, and the shortage or oversupply of skilled workers in urban areas are factors that can influence the career prospective of young dentists also in Switzerland [2,3]. So far, little information is available about the career choices of young dentists in Switzerland [2]. Employment relationships appear to have become more popular, while a decrease in the number of single private practices a few years after graduation could be observed [2]. In fact, self-employment in private practice immediately after graduation is possible, but rarely chosen. Not only for dentists, but also for the health sector and all those involved *per se*, the expectations and ideas of the growing generation of dentists regarding future professional practice and career planning are immensely important because they directly affect the implementation and provision of oral health care throughout the country, both in urban and in rural areas [4]. In Switzerland, like in other Western European countries, buzzwords such as the feminization of the dental profession [5], difficulties in finding a first job after graduation [6], and mercantilization, especially due to increasingly commercial-based structures in the health sector by financial investors [3], are current topics of conversation for young colleagues.

Therefore, in collaboration with the Commission for Health Policy of the SSO and with dental schools of all Swiss universities, the dental school of the University of Bern set the goal of conducting a survey among young Swiss practitioners in order to find out their current professional activity and future prospects about job search and expectations in order to generate better support and solutions for young colleagues.

## 2. Materials and Methods

### Study Population

In 2018, the population of Switzerland was 8,544,527 (4,307,406 females and 4,237,121 males). For a total of 3.7 million private households, the life expectancy at birth was 85.4 years for women and 81.7 years for men. In 2018, according to the Federal Statistical Office of the Swiss Confederation, there were 4337 practicing dentists in Switzerland. The number of dentists per 100,000 inhabitants has hardly increased since the 1990s at the rate of 51/100,000 [7]. This observational cross-sectional study was conducted in the form of online self-administered questionnaire. All young dentists, members of the SSO, and those from the mailing lists of dental schools of all Swiss universities were invited to participate voluntarily. The survey included 1920 practitioners (born after 1978). The online website included brief information about the aim of the survey and could be reached via a link that was sent to the members by e-mail. The online website was an online questioning tool that allowed the survey participants to fill in the questions with answer categories directly on the computer (https://survey.sso.ch/gpk_sektionen). A personal survey code was included in the e-mail that was sent, which had to be entered when filling out the email. After entering the code, the user was taken to the input interface of the online survey, where the information could be entered directly on the computer. The SSO has assured that all information provided will be kept strictly confidential and will only be used for statistical purposes in a fully coded and anonymous form. The questionnaire was developed based on similar questionnaires present in literature [8,9,10,11]. The final questionnaire consisted of 20 questions distributed over three sections. The first section collected participants’ demographic data including age, gender, level of qualification, and place of work.

The second section collected information on employment (working conditions, job searching in different geographical areas, current employment correspond expectations, number of employers, level of employment, employment level meet requirements). The third section aggregated information on future perspectives (type of dental cabinet, employee or employer, full-time part-time job, career prospects). The Commission for Health Policy, together with the media service department of the SSO, pretested the questionnaire before sending it out.

Statistical analyses were performed using STATA 16 (Stata corporation, College Station, TX, USA). Frequency distribution was calculated for qualitative variables, and means and standard deviations were determined for quantitative variables. The chi-square test was used to compare study responses between questionnaire items. Fisher’s exact test was run in case that a cell count was under 5. A multinomial logistic regression model was run using as dependent variables career prospects to evaluate the association with other questionnaire items. The significance level was set at *p* < 0.05. The raw data are available as Appendix A (Row data: Row-data.xls.).

## 3. Results

The survey included 1920 practitioners, of which 440 (22.92%) responded (37.05% males and 62.95% females). Almost three-quarters of the responders were SSO members (76.59%), 15.91% were students, and only 7.50% were non-SSO members. More than 90% of the subjects studied in Switzerland (92.50%).

Half of the responders (51.36%) worked in small to medium towns (between 2000/50,000 inhabitants) and at less than 30 km from the living areas (47.73%). Two-thirds of the sample declared to be employed by one employer, and only 2.26% by more than two employers, the majority of which (50.50%) with a part-time contract of less than 50%. The satisfaction about the working conditions ranged among positive (39.77%), rather positive (11.36%), and negative (39.32%). Almost the same number of dentists reported that they would like to work in an individual practice (*n* = 173; 39.32%) or in a joint practice (*n* = 175; 39.77%). The personal career prospects were reported as negative in 16.82% of the sample.

The majority of dentists surveyed most often saw their career prospects as neutral (39.77%) or rather positive (39.32%). Female dentists most often gave a rather positive view and male dentists a predominantly neutral view (*p* < 0.01). Significantly fewer dentists saw their professional future in a negative light (16.82%), and the fewest dentists of both sexes (4.09%) saw their career prospects as positive by far (Table 1).

The participants most often viewed their career prospects as positive (31.59%) or neutral (29.09%). It is noticeable and statistically significant (χ^2^_(6)_ = 14.17, *p* = 0.03) that a large proportion of the survey participants saw their career prospects as negative (16.82%) and the fewest as positive (4.09%) (data not in table).

Working localities were quite balanced among small, medium and large cities (26.59%, 27.50%, and 35.91%), and only villages were significantly less represented (*p* = 0.03). Few participants saw their future negatively and the fewest positive (Table 2).

Most of the participants in the study had parents with a dental education (80.92%), and only 19.8% did not. Of the dentists with a dental family background, most of them had neutral (34.25%) or rather positive (29.66%) career prospects, significantly less negative (14.25%), and the fewest positive (2.73%). Dentists with no family background were most likely to see their career prospects as positive (9.89%) and neutral (5.98%), very few negative (2.53%), and only 0.68% positive (Table 3).

The place of work (Table 4) played the most prominent factor to evaluate the career prospective positive (Coeff = −1.09 *p* < 0.01). Otherwise, gender was the most prominent factor for a rather positive career prospect (Coeff = −0.63 *p* = 0.03) with respect to a negative prospective.

## 4. Discussion

The ongoing changes of the profession often raise issues on the political agenda to develop solution strategies to support decisionmakers. To the authors’ knowledge, this was the first study in Switzerland to date investigating the young dentist career prospects. The goal of this study was to gain information about the current professional situation immediately after graduation and identify the deadlocks and challenges for professional start, current employment, and future prospects through an online questionnaire.

The results of this survey provide relevant information that is essential for professional, targeted and sustainable oral health care for the Swiss population. To date, the careers of young professionals in Western countries could be discerned usually by reports of establishment of sale of practices and registers of employed professionals. Young dentists in Switzerland most often view their career prospects rather positive or in a neutral prospect. Despite the increasing establishment of dental centers with predominantly salaried dentists, the independent individual private practice is still the most common form of practice in Switzerland [2] and also in Europe [1].

In the current survey, a large portion of dentists stated that they would like to work in an individual or in a group practice. A private practice is perceived as way of working with professional and clinical freedom, financial reward, and the possibility to spend enough time with patients [12]. Otherwise, the National Health Service in the United Kingdom (UK) is perceived as a working environment associated with the opportunity to gain clinical experience and the chance to access to specialist training. The two styles of work are often in contrast, both with strengths and weaknesses. Young dentists in UK reported to prefer to work in private practice [13], while a similar sample of Brazilian dental students at the end of their academic career reported the opposite [14]. This dissimilarity preference may be influenced by characteristics associated with the work, such as financial lucrativeness, professional status, job security, flexible work options, and the independence of a career in healthcare [12,15,16,17,18]. The longing for a good work-life balance is strongly related to personal life in terms of long-term career expectations [12,16]. The above-mentioned advantages appear to have a direct influence on the choice of dental profession [15]. The personal and career gratification has been reported as the most important single factor [8]. The possibility to determine their own proper working hours and be their own boss was very important to almost half of all respondents in an Indian study [19,20]. The majority of the participants in the present study reported to have family members working in the dental area, showing that the strongest influence on career choice in dentistry are the parents or family. In terms of career goals, the most common short-term career goal was to achieve financial stability and to pursue professional training [19]. The short-term career expectations of final-year students were to develop skills and competences and to gain experience and additional knowledge. About two-thirds of the respondents stated that they wanted to work full-time, with more male than female dental students. Childcare obligations were main reason for female dentists to transition from full-time to part-time work [17,21]. Otherwise, in the present paper, no gender effect was stated. It is possible that this issue is limited to Western society, where the paper was designed and performed. Even if it is necessary to enlighten that, the present survey did not explicitly target the relationship between work and family life or children, respectively.

Although gender differences were not explicitly focused on in the present study, there are patients who prefer treatment by both male and female dentists [22]. Furthermore, female dentists account for more than 60% of diplomas in Switzerland [2]. These issues mean that a general discussion about the changing in the choice of professional practice, as well as the degree of time spent working, is needed by professional organizations. Especially with regard to the calculation of oral health care needs, it is necessary to take such factors into account for policymakers and those responsible for professional organization to obtain information. It is noteworthy that the feeling of job security decreased significantly during the academic time span and after some years working [18], probably due to pressure that professionals are exposed to and the uncertain future. The concern about the lack of funding from the National Health Service in the UK [18] is not comparable to the private Swiss system with the free choice of doctor and therapy by the patient. The future employment prospect rate for the future is also a factor creating concern in the UK [18,23], as well as employment prospects and lack of employment in the following year [18]. Differences in the career plans of dental students with different backgrounds are due to socioeconomic backgrounds [24].

In this first study on the future prospects of young dentists, information was obtained that is important for the political and economic design of oral health care in Switzerland. The expectations of the young colleagues are therefore one of the primary goals and are absolutely necessary for the shaping of the oral health care landscape, as well as for advising the political decisionmakers and ministries. The desire for the type and scope of dental work, whether employed or self-employed, is directly decisive for oral health care. Although the employment of dentists was asked about in this study, no causality is possible, as no reasons were given for this. Working conditions and differences between the different types of practices, especially with regard to work-life balance, should also be examined in more detail. Also, the frequently discussed problem that dental centers and private dental practices make different therapy decisions for the same findings, as well as a possible influence of a financial investor compared to a self-employed or employed dentist on the therapy carried out, was not covered [25]. In addition, the COVID-19 crisis currently represents a global health and economic challenge. This can also change the oral health landscape massively. Additional protective measures, new practice concepts, a lack of patient flows, and patient restraint, both through social distancing and cost-intensive treatments, have an impact on the professional/economic situation. The changeover to short-time work and the lack of renewal of fixed-term employment contracts, as well as dismissal of employees, became necessary in some dental practices in Switzerland in order to maintain the necessary liquidity of the practice to be able to continue to work and care for patients. Unfortunately, these aspects are not reflected in the current study and should therefore be investigated further. A follow-up investigation of the investigated target group also appears to be very useful to see whether the expectations have been fulfilled or whether the professional situation of the investigated target group has changed. Although the number of participants appears to be large in relation to the total number of dentists in Switzerland, another weak point of the study may be that not all dentists in Switzerland are covered by the Federal Statistical Office. In spite of constant immigration and emigration of dentists and recognition of dentist diplomas in Switzerland, no data are available on the total number of dentists in Switzerland. Therefore, only the numbers of the members of the SSO can be taken as a reference value. The number of dentists practicing in Switzerland is therefore likely to be higher. The significance of such investigations with regard to the situation of dentists, either to be employed or self-employed, has again come closer into focus, not only because of the COVID-19 crisis. The need for information and support for career choice is immense and should be investigated more closely.

## 5. Conclusions

Within the limitations of the current study, it can be concluded that young and aspiring Swiss dentists are satisfied with their career prospects. The reasons for this need further investigation. There is a desire for cantonal practice assistance and mentoring programs, as well as support in finding a job and in making the leap into self-employment.

## Figures and Tables

**Table 1 ijerph-17-04310-t001:** Distribution of the population between gender and career prospects.

Gender	Positive *n (%)*	Rather Positive *n (%)*	Neutral *n (%)*	Negative *n (%)*	Total *n (%)*
Males	6 (1.36)	81 (18.41)	53 (12.05)	23 (5.23)	163 (37.05)
Females	12 (2.73)	92 (20.91)	122 (27.73)	51 (11.59)	277 (62.95)
Total	18 (4.09)	173 (39.32)	175 (39.77)	74 (16.82)	440 100.00)

Pearson χ^2^_(3)_ = 11.75, *p* < 0.01.

**Table 2 ijerph-17-04310-t002:** Distribution (number of participants and percentage) of the association between future career prospects and working location.

Working Location	Positive *n (%)*	Rather Positive *n (%)*	Neutral *n (%)*	Negative *n (%)*	Total *n (%)*
Village	8 (1.82)	9 (2.05)	24 (5.45)	3 (0.68)	44 (10.00)
Small City	5 (1.14)	53 (12.05)	44 (10.00)	15 (3.41)	117 (26.59)
Medium City	3 (0.68)	45 (10.23)	45 (10.23)	28 (6.36)	121 (27.50)
Large City	2 (0.45)	66 (15.00)	62 (14.09)	28 (6.36)	158 (35.91
Total	18 (4.09)	173 (39.32)	175 (39.77)	74 (16.82)	440 (100.00)

Pearsonχ ^2^_(6)_ = 14.17, *p* = 0.03.

**Table 3 ijerph-17-04310-t003:** Association between parents with dental education and career prospects. Distribution (number of participants and percentage) of the association between future career prospects and parents with dental education.

Parents with Dental Education	Positive *n (%)*	Rather Positive *n (%)*	Neutral *n (%)*	Negative *n (%)*	Total *n (%)*
No	3 (0.68)	43 (9.89)	26 (5.98)	11(2.53)	83 (19.08)
Yes	12 (2.73)	129 (29.66)	149 (34.25)	62 (14.25)	352 (80.92)
Total	15 (3.45)	172 (39.54)	175 (40.23)	73 (16.78)	435 (100.00)

Fisher’s exact test *p* = 0.08 (not responders = 5).

**Table 4 ijerph-17-04310-t004:** Multinomial regression analysis. Career Prospects (Positive/Rather Positive/Rather positive/Negative). Negative prospect as base outcome.

Careers Prospective	Coefficient (Std. Err)	*p*-Value	95% CI
Negative as base outcome			
Positive			
Gender	−0.49 (0.70)	0.48	−0.88/1.87
Place of Work	−1.09 (0.33)	<0.01	−1.74/−0.44
Parent/Dentist	−0.62 (0.74)	0.40	−2.08/0.83
Age	0.38 (0.36)	0.28	−0.32/1.10
Constant	0.26 (1.63)	0.84	−2.94/3.46
Rather positive			
Gender	−0.63 (0.30)	0.03	−1.22/−0.05
Place of Work	−0.22 (0.15)	0.14	−0.51/0.07
Parent/Dentist	−0.66 (0.38)	0.07	−1.40/0.07
Age	0.15 (0.17)	0.37	−0.18/0.50
Constant	2.62 (0.85)	0.02	0.94/4.29
Rather negative			
Gender	0.01 (0.38)	0.92	−0.59/0.61
Place of Work	−0.27 (0.15)	0.06	−0.55/0.01
Parent/Dentist	−0.04 (0.39)	0.91	−0.82/0.73
Age	−0.11 (0.18)	0.50	−0.46/0.22
Constant	1.69 (0.88)	0.05	−0.03/3.42

Number of observations = 440, χ^2^_(12)_ = 36.97, Log likelihood = −481.26, *p* = 0.04.

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
