# Peer review of "Career Prospects of Young Dentists in Switzerland"

_ijerph, 2020, doi:10.3390/ijerph17124310_

Round 1

Reviewer 1 Report

How did the authors evaluate the non-members of SSO Swiss Dental, if the survey was sent to members of the SSO Swiss Dental Association?

Why did not they evaluate in the survey the interest in speciality studies or speciality that they studied?

Is it correct to discuss the importance of specialization in the India when the authors did not evaluate the specialities studies in Switzerland?

I recommend to authors to use the Running title as a Title, that is more related with the manuscript.

Overall comments

This study is interesting and it reflects the different aspects of dentistry studies. This manuscript describes, in general, the point of view of dentistry studies in Switzerland, the authors conclude adequately the limitation of the study. However, I have some corrections that authors need to resolve. Thus, I recommend this manuscript with minor changes.

Reviewer 2 Report

A survey on the self assessed career opions of young dental professionals certainly is of interest.

However, there are severeal issues that need to be addressed.

Introduction:

1. While the relevant literature is cited in the discussion, the information given in the introduction is far less comprehensive.

2. Numbers of 1,000 plus should be separated with a comma as decimal space. Only some of the numbers in this manuscript follow this rule.

3. Line 7, page 2: Please rephrase "The proportion of women in the federal diplomas awarded anually..." The meaning is not fully clear. Did you mean: " The proportion of women among those awarded with a federal diploma..."? Every year, this year, a certain year?

4. Line 15: ...while a decrease of the number... there is a verb missing in this part of the sentence.

Materials and methods

5. Study Population, line 6/7: The population is not clearly defined. First, "all members of the SSO", then "1,920 practitioners born after 1978", later on students are mentioned to have responded and that also non-members of the SSO have answered the questionnaire. Please clarify, e.g. how exactly the choice was made whom to invite and who exactly participated.

6. line 5 of this paragraph: "The number...has hardly increased since...the rate 51/100,00." Please rephrase this sentence in order to clarify its meaning.

7. 4th to last line page 2: Was there any pretesting of the questionnaire? Please describe.

8. next line: questionnaires - plural, please add s

9. Page 3, 2nd paragraph, 3rd line: "Chi-square test..." Word order. Either start or  end with  the if-clause, but do not put it in the  middle.

Results

10. 2nd paragraph: "Half of the...work..."  Delete s

11. small-medium: please clarify "small or medium" or "small to medium"?

12. 2nd paragraph, line 6: "reported to" probably should read "reported they"

13.  3rd paragraph, line 4: "including more questions than men" - meaning unclear, please rephrase.

14. 4th paragraph, line 2: statistically significant - delete "ly" from "significant" and delete "," behind "...=0.03)" as this is a necessary clause.

15. 5th paragraph: "Fewer participants..." compared to? Should this read "Few"?

16. Page 4, end of first paragraph: There is a "rather" missing at one of the "positive"

17. 2nd paragraph: "Vision  of career rather...respect to a negative..." Please rephrase the sentence in order to clarify the meaning.

18. Obviously, not all results of the questionnaire are presented in this paper. Which results were chosen, which not and why?

Discussion:

19. There are quite a few spelling errors throughout the discussion. Please proof-read thoroughly.

20. How do your assumptions on the Impact of the COVID-19 crisis realte to your results?

21. "...dental centers and private dental practices make different therapy decisions for the same findings" On what Facts is this Statement based?

Reviewer 3 Report

The manuscript provides quite interesting data on the perspectives of dentists and their future prospects. A diverse panel of information is provided, which helps build a rich environment. The manuscript is appropriate, though it requires minor editing.

The level of English can be improved, as certain ambiguities and grammatical issues exist, for example, statements such as 'proportion of women in the federal diplomas'. Furthermore, captions can be revised to be clearer.

Please use a thousand indicators consistently, as it is often either 1000, 1'000, 1,000, 1.000, or 1 000.

If I gather correctly, the questionnaire was distributed by the SSO, however, a proportion of the respondents were not members thereof. Is this due to adjunct distribution?

Were the questionnaires validated prior to distribution?

Was the study approved by an ethics review board?

For the statistical analysis, why express the values relative to the total cohort, instead of per gender?

The paragraph above and below Table 1 are slightly confusing. What is the difference here as both speak to the positivity, or are these different contexts? The second paragraph refers to data not in the table, but the context thereof is confusing, and its unclear why it is not included.
